# Language Perceptions of New Mexico: A Focus on the NM Borderland

Kathryn P. Bove

Department of Languages and Linguistics, New Mexico State University, Las Cruces, NM 88003, USA;
kpbove@nmsu.edu

**Abstract:** New Mexico is located along the U.S.–Mexico border, and as such, Spanish, English, and language mixing form an integral part of the New Mexican identity. New Mexico is often divided into a northern and a southern region with the north known for Spanish archaisms due to historic isolation, and the south associated with ties to a Mexican identity due to the location of the U.S.–Mexico border. The current study uses perceptual dialectology to capture the way in which speakers in the south of New Mexico perceive this north/south divide and communicate their identity. Overall, there is evidence of the north/south divide, but speakers in southern New Mexico focus much more on language use such as Spanglish, English, and Spanish than on their northern counterparts. Participants reference language mixing over language "purity" and borders over an explicit rural/urban divide. Like previous accounts, we see reference to the "correctness" of both English and Spanish, examples of specific terminology used in different parts of the state, and descriptions of accents throughout the state.

**Keywords:** perceptual dialectology; mapping; New Mexico; border





## 1. Introduction

Language is an important part of the New Mexican identity. While the concept of New Mexico Spanish may appear to be singular, there is a significant body of research that suggests there are at least two unique speaker groups in this geographic area: Northern New Mexico Spanish, and Southern New Mexico Spanish. In their work on Spanish in New Mexico and Southern Colorado, Bills and Vigil (2008) describe a spatial divide between the northern part of New Mexico, which presents many unique characteristics due to the historic isolation of Spanish, and the southern part of state, whose identity is much more closely tied to the border and the transfronterizo life, in which residents of this area cross the border regularly for work, recreation, family, etc. This distinction is well documented by linguists, but the current study replicates the study by Vergara Wilson and Koops (2020/2015) and uses perceptual dialectology (introduced by Preston 1989) to see how speakers in the southern part of New Mexico view themselves and others within the state.

Perceptual dialectology is a subfield of sociolinguistics that researches what nonlinguists think about linguistic practices, language history, and perceptions of variation. In recent years, it has been used to study language in individual areas within the United States, including Kentucky (Cramer 2013), Tennessee (Fridland and Bartlett 2006), Ohio (Benson 2003), Washington (Evans 2011), California (Bucholtz et al. 2007), and Miami (Callesano 2020), to name a few. Recent work by Vergara Wilson and Koops (2020/2015) uses perceptual dialectology to study bilingualism and language attitudes in New Mexico, focusing on the use of Spanish. Their participants, who all currently reside in northern New Mexico, note ideas of language 'purity', unique features of Spanish of the north, and the border life of the south, an idea which supports Bills and Vigil's (2008) spatial divide. Like previous studies mentioned above, there is also a discussion of the perception of a rural/urban divide within the state. The current study recreates Vergara Wilson and Koops'

(2020/2015) study with speakers from the southern part of the state. Overall, the current research will compare the perceptions of speakers in the north (from Vergara Wilson and Koops 2020/2015) with speakers from the south (current study). As such, this article aims to answer the following research questions:

1. To what extend does the north/south and rural/urban divides observed in previous work also appear in the current data set?
2. How do southern New Mexicans perceive their language, and to what extent does this vary across the state?
3. What language do participants use to divide language groups (i.e., words, prosody, food, descriptions of people, etc.)?

This paper is organized in the following way: Section 2 reviews the previous literature on language attitudes, perception mapping, and previous accounts of language attitudes in the U.S. Section 3 reviews the methodology, and Section 4 presents the results and discussion. Lastly, there is a brief conclusion in Section 5.

## 2. Literature Review

### 2.1. Folk Linguistics and Perceptual Dialectology

Folk linguistics uses ideas of language from non-linguist speakers to give linguists a clearer idea of how speakers view themselves, which can be used as a starting point for new research. When it was introduced, Hoenigswald (1966, p. 20) stated the importance of this subfield within the larger field of sociolinguistics:

> 'we should be interested not only in (a) what goes on (language), but also in (b) how people react to what goes on (they are persuaded, they are put off, etc.) and in (c) what people say goes on (talk concerning language). It will not do to dismiss these secondary and tertiary modes of conduct merely as sources of error.'

Since its introduction, the use of folk linguistics in academic work has been met with some backlash from the sociolinguistic community (see Niedzielski and Preston 2000 for a detailed description). However, there has been consistent support for this subfield for decades. For example, Albury (2017) argues that folk linguistics can make significant contributions to critical sociolinguistics because it gives a voice and legitimizes communities and underrepresented language. Like the current study, Martinez (2003) uses folk linguistics to highlight the changes in language that occur along the United States–Mexico border. He argues for this methodology by stating that it 'can shed light on how those on the border construct dialect perceptions and the social values that underpin these constructions' (Martinez 2003, p. 39).

### 2.2. Perception Mapping

Preston (1999) describes perceptual dialectology as a 'sub-branch' of folk linguistics. Perceptual dialectology aims to draw dialect boundaries similar to dialectology (e.g., Chambers and Trudgill 1998), but the important distinction is that perceptual dialectology does so based on non-linguist perspectives. Within this methodology, participants are asked to complete at least one of the following tasks: draw dialect boundaries; label dialect regions; describe dialect regions; or identify a degree of difference, labeling a dialect as "correct" or "pleasant". More recent additions to the methodology include dialect identification (Cukor-Avila et al. 2012 in Texas), matched guise tests to measure language attitudes (Garrett et al. 2003 in Wales), naming dialects (Alfaraz 2002 in Miami), imitation (Adank et al. 2013 in Glasgow), and term frequency using social media and heat maps (Garzon 2017 and Callesano 2020 in Miami). Preston (2018) also argues for benefits of using technology such as GIS in modern perceptual dialectology.

The current study uses a draw-a-map task. Most of this type of perceptual dialectology can be traced back to Preston (1989). Within this methodology, participants are given little instruction other than to indicate how people speak and where. In general, participants tend to first draw stigmatized areas (Preston 1999) and the areas where they consider themselves

local (Gould and White 1986). In his early work, Preston (1982) had students from the University of Hawaii draw on maps to indicate how people speak. According to Niedzielski and Preston (2000, p. 46), the first U.S. maps were blank, and this caused confusion for participants. Therefore, researchers began using U.S. maps with state lines. Preston's (1986) study compared responses from participants in multiple locations including Hawaii, southern Indiana, western New York, New York City, and southeastern Michigan, and the research discussed correctness, pleasantness, degree of difference, and placement of regional voices. An example of the maps is shown in Figure 1 below (Niedzielski and Preston 2000, p. 47):

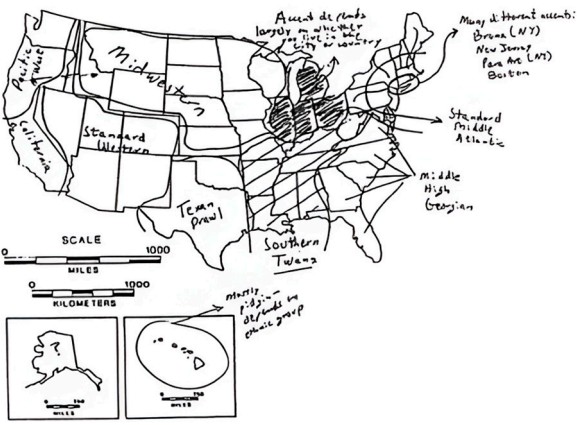

**Figure 1.** Map from Hawaii participant.

In this study, one example of a question asked was "Where is the core of the 'south'?", which many placed in Alabama (Niedzielski and Preston 2000, p. 57). While there was no significant findings related to gender, Niedzielski and Preston (2000, p. 57) discuss the connection between perceptions and social status observed, stating that participants often used terms like "standard" or "normal" versus "high-falutin", "very distinguished", or "snobby".

While the previously mentioned studies analyze maps of the United States in its entirety, there is a group of research that focuses on individual states and the perception of that state's residents. From this body of research, there are several themes that can be observed. First, there is a distinction between the perception of rural and urban speakers, in which speakers from urban areas are often rated as more "correct" than speakers from rural areas. For example, Fridland and Bartlett (2006) replicate Preston's (1989) study in Memphis, Tennessee, to measure both correctness and pleasantness. Speakers in Memphis rate themselves differently, and they often judge those in the south as "less correct", which highlights this rural/urban divide. However, there was no difference in pleasantness between rural and urban areas, which suggests that there is not a direct connection between correctness and pleasantness. In Kentucky, Cramer (2013) focuses on identity formation through enregisterment, iconization, and recursivity. Louisville is located on both a geographic border, the Ohio River, and political border between Indiana and Kentucky. She finds that this border is a creator of identity for participants. Speakers in Louisville also separate this city from the rest of the state, highlighting a similar rural/urban divide as observed in Fridland and Bartlett's (2006) Tennessee work.

Another theme highlighted in this body of research is "differentness" of language. Benson (2003) studies perceptual dialectology through maps with residents from Ohio. In addition to the map drawing task, she measures the degree of differentness to distinguish dialect boundaries. Benson (2003) found that speakers in central and northwestern Ohio communicate a unique identity that distinguishes them from other parts in Ohio. Speakers from southeast/central and southern Ohio communicate the idea of one unified way to speak in Ohio, which Benson concludes is a move that stems from speaker linguistic

insecurity. Similarly, Evans (2013) asked participants to indicate language differences in Washington state. Based on the data, there were minimal distinctions made in the way in which Washintonians speak, leading Evans to conclude there is a perceived homogeneity in speech. She does note a rural/urban divide that some participants observed, specifically noting population size as a factor in the way people speak. Evans also notes the importance of outlier data in this style of perceptual data collection.

Lastly, this research frequently discusses lexical trends that participants recognize when mapping dialects. Bucholtz et al. (2007) uses data on language perception of California by Californians. Student researchers collected map data that informed geographic regions/labels, social and linguistic labels, language and dialect labels, slang and other lexical labels, and social group and attitude labels. Figure 2 is an example of a map taken from the study:

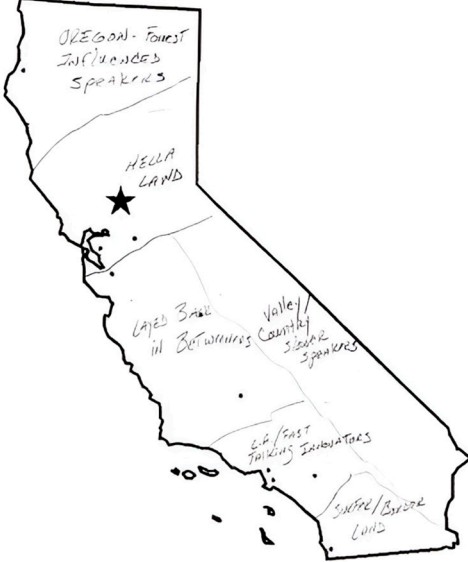

**Figure 2.** Participant California map, Bucholtz et al. (2007, p. 331).

The lexical lists provided in each category identified by Bucholtz et al. (2007) create a perceived profile for the speaker groups of California that, as seen in Cramer's work (2013), build a perceived identity of the speakers. Like other studies, Bucholtz et al. (2007) use "best" and "worst" in their research to encourage participant evaluation of Californian dialects. In their discussion, they recognize that the use of these terms may be implicitly endorsing a standard language ideology.

*2.3. The Research Area: Southern New Mexico*

New Mexico is located in the southwestern U.S. Like other southwestern states, there were several groups of Native peoples in this area. After the arrival of the Spanish and later Mexican independence, New Mexico became a U.S. state in 1912. Despite the change in the political border, there are still ties to both Spanish and Mexican culture and language to this day. There is a common myth that the Spanish spoken in northern New Mexico reflects sixteenth-century Spanish. Bills and Vigil (2008, p. 14) state that:

> "the reality of New Mexican Spanish is much more complete and quite different from a magical association with Spain. That reality is accurately reflected in the everyday labels that speakers of New Mexican Spanish ordinarily employ to describe their ethnic and linguistic identity: *somo mexicanos* 'we are Mexican' *hablamos mexicano* 'we speak Mexican'".

New Mexican Spanish and the border with Mexico are and have been an integral part of the identity of the state of New Mexico, especially in the southern part of the state. According to the United States Census Bureau (2019), just over a quarter of New Mexico

residents speak Spanish at home, and language mixing or *Spanglish* can be frequently observed throughout the state. Spanish has been adapted into the English of this area so strongly that it would be impossible to communicate without it. Historically, Waltermire (2017, p. 179) identifies three major factors shaping New Mexico's English and Spanish use: (1) relative isolation from other Spanish speaking populations; (2) the gradual settlement of English speakers beginning in the mid 1800s; and (3) waves of immigrants during the second half of the twentieth century. New Mexico is the fifth largest state by area, but it has a small population of just over two million residents (United States Census Bureau 2019). New Mexico shares state borders with Arizona, Colorado, and Texas to the west, north, and east, respectively, and importantly, it shares an international border with Mexico to the south.

In the southern part of New Mexico, language use continues changing rapidly. Waltermire and Valtierrez (2019, p. 414) describe the current language situation in the Las Cruces area as the greatest concentration of Spanish speakers in southern New Mexico, with immigration and migrant farm workers contributing to the Spanish spoken in this area and the preservation of the language. Many studies on the grammatical features of New Mexico Spanish (see, for example, Cisneros et al. 2023) recognize the quickly changing Spanish spoken in New Mexico. Overall, we can observe three salient features of New Mexican Spanish: the abundant code-switched utterances (Bills and Vigil 2008; Wilson and Martínez 2011), archaisms and anglicisms (Bills and Vigil 2008; Waltermire 2017), and borrowings (Waltermire and Valtierrez 2019). Nowadays, the use of Mexican Spanish has traditionally been fairly stigmatized (see Waltermire 2017, pp. 181–82, for comments by New Mexican Spanish speakers).

The current study builds on a study of the perceptual dialectology of New Mexico by Vergara Wilson and Koops (2020/2015). In this study, the authors aim to discover how New Mexicans (1) construe language variation in their own state in spatial terms as well as (2) communicate correctness and prestige of language in this state. Participants were given a map of New Mexico and asked to indicate how New Mexicans speak, and in a questionnaire, they were asked where people speak the "best" and the "worst". Overall, the following are the two most notable patterns that emerged: First, Bills and Vigil (2008) observe a north/south divide in the language of New Mexico in which speakers and linguists alike recognize key differences between these two parts of the state. In their draw-a-map task, Vergara Wilson and Koops (2020/2015) observe that speakers recognize north and south in both their map drawing task and their discussion of "best/worst" language. The second notable finding was the urban/rural divide. Vergara Wilson and Koops (2020/2015) find an important distinction between rural and urban areas. As Vergara Wilson and Koops (2020/2015) point out, much of the state of New Mexico is rural. When discussing the "best" and "worst" Spanish, participants indicated that they perceived the "worst" Spanish is spoken in rural locations and "better" Spanish is spoken in more urban locations. The current study will replicate this study to investigate these themes using participants from the southern part of the state.

## 3. Methods

Student research assistants collected data from 314 participants during the fall of 2019. All participants are current residents of southern New Mexico, and they have spent the majority of their lives living here. First, each participant was given a "draw-a-map task" (Preston 1989), a methodology originally borrowed from cultural geographers (e.g., Gould and White 1986 and seen in many perceptional dialectology studies, including those of Vergara Wilson and Koops (2020/2015). As with Vergara Wilson and Koops (2020/2015), the participants were given the option to use Spanish or English, and many often code switched within the survey tool. An example of the map is presented in Figure 3.

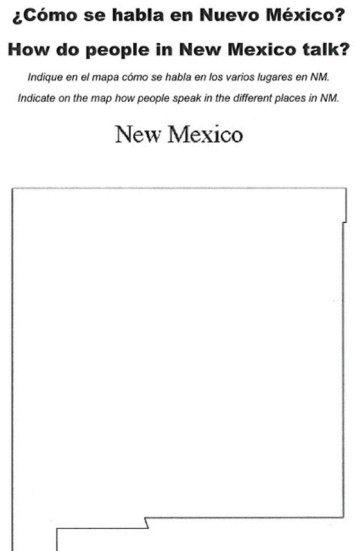

**Figure 3.** The New Mexico map task.

First, participants were given the blank maps and only given the instructions to "indicate how people speak in the different parts of New Mexico". On the reverse side of the map, participants filled out a survey in which they were also asked about their own language use and preferences. This included participant information such as age, gender, and place of birth, as well as language use questions such as which language is prefered in different social situations. Lastly, in an effort to replicate the study carried out by Vergara Wilson and Koops (2020/2015), I wanted to ask questions regarding the "best" and "worst" language, but as previous studies like Bucholtz et al. (2007) recognize that the use of these terms may be implicitly endorsing a standard language ideology, the questions were reworded to ask "In your opinion, in which part of the state do you like the way people speak? Why?". This is a slight variation of the "best"/"worst" question used in previous research. Incomplete survey tools were removed, leaving us with 314 maps to be analyzed.

The current study focuses primarily on the data provided on the maps by the participants. In the analysis of these maps, there were several considerations. First, the number, placement, and shape of any boundaries such as lines, as well as discussion of borders, cities, and natural landmarks was documented. Second, I noted descriptions of language, which included reference to specific languages, descriptions of dialects, or other language judgements. Additionally, based on Vergara Wilson and Koops' work (2020/2015), I noted reference to "north/south" and descriptions of rural/urban areas. Finally, lexical descriptions were collected. I follow the work of Bucholtz et al. (2007) in using the following labels to categorize lexical descriptions of speakers: geographic regions/labels, social and linguistic labels, language and dialect labels, slang and other lexical labels, and social group and attitude labels. For the current study, only the "best/worst" questions were considered and used to support the data collected from the maps.

## 4. Discussion of Results

Based on the analysis by Vergara Wilson and Koops (2020/2015), the results begin with a discussion of the north/south and rural/urban divides, as well as additional geographic classifications observed in the current data, such as the importance of borders. Next, the discussion focuses on the languages identified in the maps, as well as the role and judgements of language mixing. Lastly, we review lexical categories used to describe both the languages and their speakers on the maps.

*4.1. North/South Divide and Further Devisions*

Following Bills and Vigil's (2008) north/south divide, Vergara Wilson and Koops (2020/2015) found extensive use of this distinction, in particular when discussing the "worst" and "best" languages. The current study also found evidence of the north/south divide, which was evident in both maps and qualitative answers.

Figure 4, like many of the maps, shows a single line separating the north from the south of the state. In the north, this participant indicated the presence of Native peoples in the northwest corner, "proper" Spanish in the northeast, "country" speakers along the east, and Spanish just north of the two largest cities in the state, Albuquerque and Santa Fe, which are labeled with "slang". In the south, this participant only indicates that speakers use Spanish, English, and Spanglish. Like in Vergara Wilson and Koops' work (2020/2015), I found many references to the north and south in the answers to the question about best and worst way of talking, and many participants also differentiate between speaker groups by using reference to the north/south divide. The current study also finds that there was also distinction in the questionnaire, where participants indicating preferring how people speak "in the north" or "in the south".

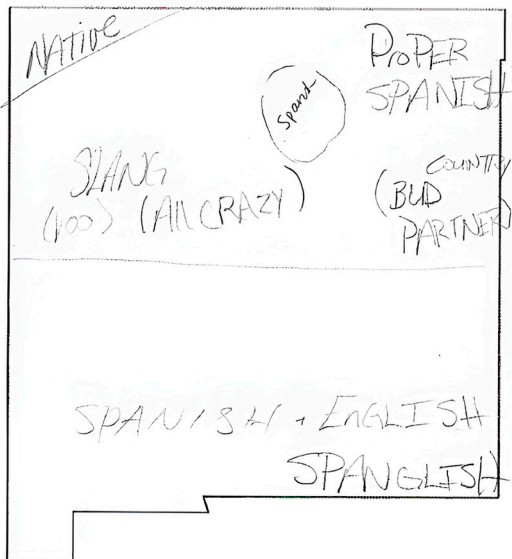

**Figure 4.** North/south divide in New Mexico Maps.

This study also finds that there was also frequent use of a 'middle zone' that was separate from both the north and the south. This third area was drawn onto maps but was not referenced in any descriptions of language use in the questionnaire.

This "central zone" is where the largest cities in New Mexico, such as Albuquerque and Santa Fe, are located, and this identification of a central zone separates this urban zone from rural northern towns known for their marked New Mexico Spanish. Figure 5a describes this central zone as an "even-ish split of Spanish and English", while Figure 5b shows more of a scalar distinction, showing that there is more Spanish in this central zone than the north, but less Spanish than in the southern zone. Both maps indicate that there is more Spanish along the southern border and more English in the north. This is a trend frequently seen in the maps.

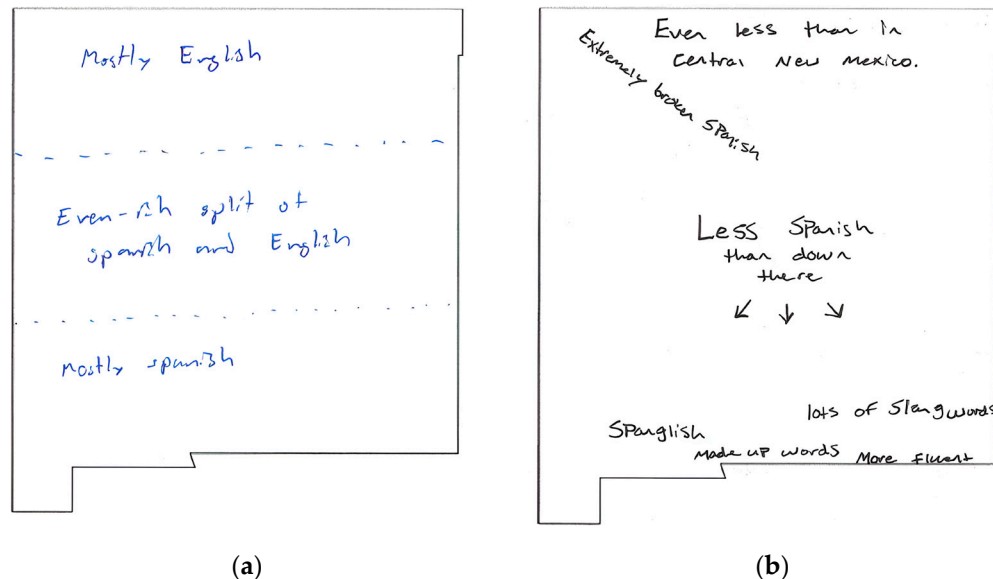

(**a**)                                                        (**b**)

**Figure 5.** (**a**,**b**) Central zone in New Mexico maps.

There were also several maps that had even more layers. These layers are mostly north/south dividing, and the factor that divides the layers was often the "correctness" of Spanish and English or the presence of mixing.

In Figure 6a, we see layers that reference correctness and language "purity", which are seen in the work of Vergara Wilson and Koops (2020/2015). This participant states that, in the north central part of the state, 'the majority of people in NM are bilingual and it is difficult to speak both languages correctly' (translation mine) and proceeds to identify a more southern layer as 'Spanglish'. It is notable that this participant differentiates bilingualism and Spanglish. In Figure 6b, the participant describes the type of Spanish and English in each layer, which ranges from "proper/formal" English in the north, "ghetto" English in the north central (where the two major cities are), "country" Spanish in the south central, and Spanish (without a dialect description) in the south.

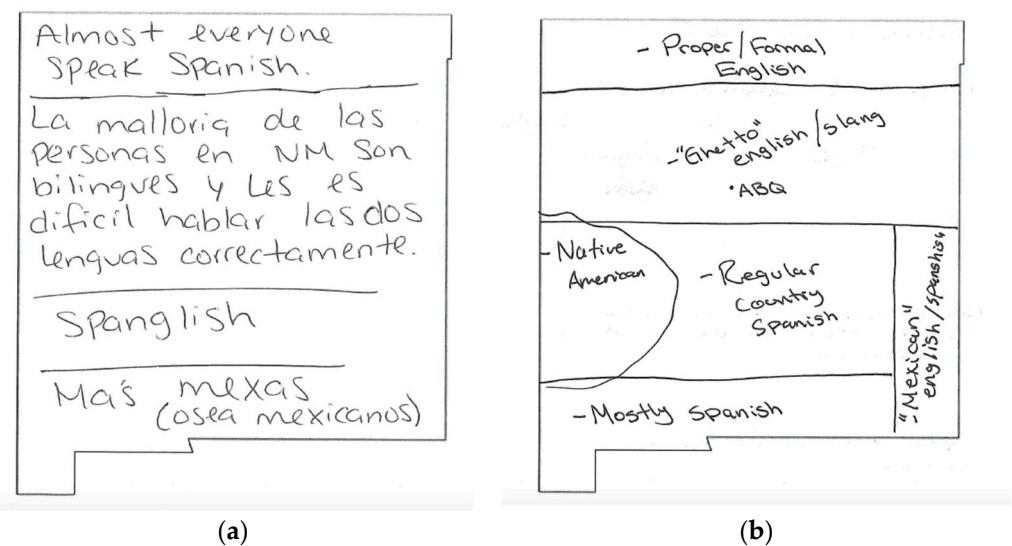

(**a**)                                                        (**b**)

**Figure 6.** (**a**,**b**) Layers in New Mexico maps.

Lastly, Vergara Wilson and Koops (Wilson and Koops 2020/2015) observed a clear divide in the description of Spanish of each area: descriptions like "colonial" were used exclusively in the north, and "Mexican" or "Mexican influence" were used exclusively in

the south. Spanglish, however, was used throughout. This idea of archaic language in the northern part of the state is something that characterizes the northern part of the state (although this may be more of a stereotype (see Waltermire 2023)). Only one participant referenced this aspect of Northern New Mexico Spanish, as seen in Figure 7.

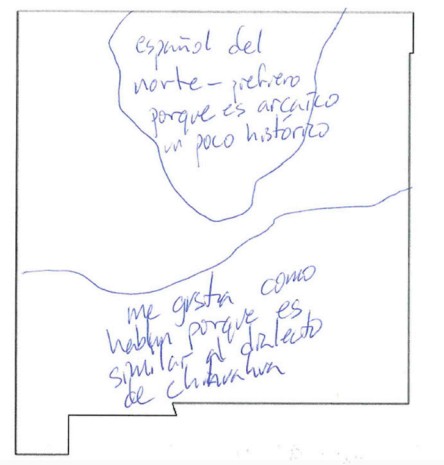

**Figure 7.** Map referencing 'archaic' dialect.

In Figure 7, the participant labels the north with "*español del norte- prefiero porque es arcaíco, un poco histórico* "northern Spanish- I prefer because it is archaic, a little historic" and the south with "*me gusta como hablan porque es similar al dialect de Chihuahua* "I like how they talk because it is similar to the dialect from Chihuahua" (translation mine). This map communicates the ideas held by the traditional literature in the Northern New Mexico vs. Southern New Mexico divide. However, no other maps in the current study make this distinction.

*4.2. The Role of Borders*

Many of the maps referenced the U.S.–Mexico border, frequently making reference to the Spanish sounding like the Spanish spoken in Chihuahua, Mexico. This reference was expected as the border is such a big part of life and identity in the southern part of New Mexico. Surprisingly, there were frequent references to neighboring states as well, such as:

(1)　'Dropping sounds, midwestern along CO border; Chicano and country along the southern border, 'Texas talk' along border with TX'.

The idea that the Colorado border hosted 'better', 'proper' or 'midwestern' English was seen in several maps. This follows Niedzielski and Preston's (2000, p. 98) observation that midwestern speech is perceived as the norm. Participants also drew additional separations as seen in Figure 6, identifying the Native area in the northwest of the state or 'cowboy' area along the eastern border with Texas, as seen in Figure 8:

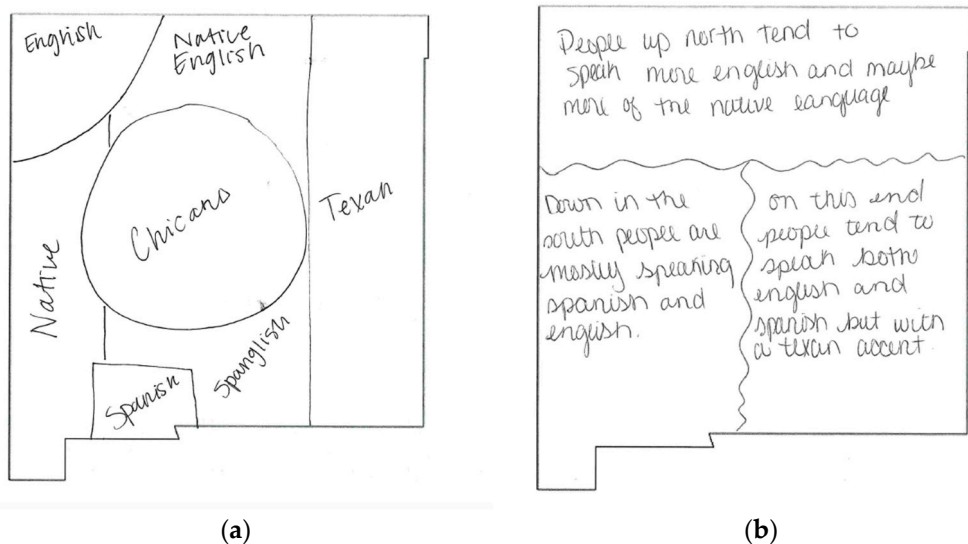

**Figure 8.** (**a**,**b**) Texas Border on map.

The participants in the work of Vergara Wilson and Koops (2020/2015) reference the U.S.–Mexico border, but we also see mention of the borders with Texas and Colorado throughout. In Figure 8a, the participant indicates that the eastern third of the state shows influence from Texas, and in Figure 8b, the participant notes the "Texan accent" that people have in the east when they speak Spanish. The recognition of borders in the creation of speaker identity has been seen in previous work, such as Cramer's (2013) argument that borders serve as a creator of identity for participants, specifically Kentucky's border and its importance in "southern" identity.

### 4.3. Rural/Urban Divide

In addition to the north/south divide, Vergara Wilson and Koops (2020/2015) highlight the importance of the rural/urban divide. As previously mentioned, participants noted the connection between the "best" Spanish spoken in urban areas and the "worst" Spanish spoken in rural areas. This is also observed in other studies such as those of Fridland and Bartlett (2006) and Cramer (2013). However, the current study did not find this explicit connection. Participants do recognize differences between rural and urban areas, as seen in the labeling of the middle layer seen in Figure 6, which suggests that participants recognize that the most highly populated areas speak differently than the more rural areas. Several participants also used lexical items in the description of individual cities that differentiate these speakers from rural speakers, as shown in below:

(2) Albuquerque: loud, aggressive, fast;
Santa Fe: slow calm Spanish;
Las Cruces: Spanish slang;
Socorro: slang, western;
Carlsbad: country English;
Hatch: country Spanish.

The cities of Albuquerque, Santa Fe, and Las Cruces are the largest three cities in the state, and there was no clear consensus that participants evaluated these locations as those that contain the best or most proper language. In fact, Albuquerque was often associated with "slang", but the use of this descriptor was not exclusive in the urban areas. The smaller towns such as Carlsbad and Hatch are associated with "country", but this is not exclusively associated with incorrectness.

In Figure 9a, the participant identifies cities in New Mexico and assigns language use to each. For example, the participant states that in Albuquerque, English, Spanish, Mandarin, and Arabic are spoken, while in Sunland Park in the south, there is exclusively

Spanish. There are additional judgments included just outside of the map to the right that indicate that in the north, language is "too formal, unnecessarily complicated" and in the south, it is "too informal, sloppy, incomprehensible". It is possible that this participant is referencing an urban/rural divide given that they drew a circle around Santa Fe and Albuquerque and references formality while describing the southern region, which includes many rural areas, as informal. In Figure 9b, the participant places "elegant" and "formal" on the large cities and uses descriptors such as "country" for more rural areas. While there is some reference to notions of rural/urban, this distinction is rarely explicitly recognized by participants.

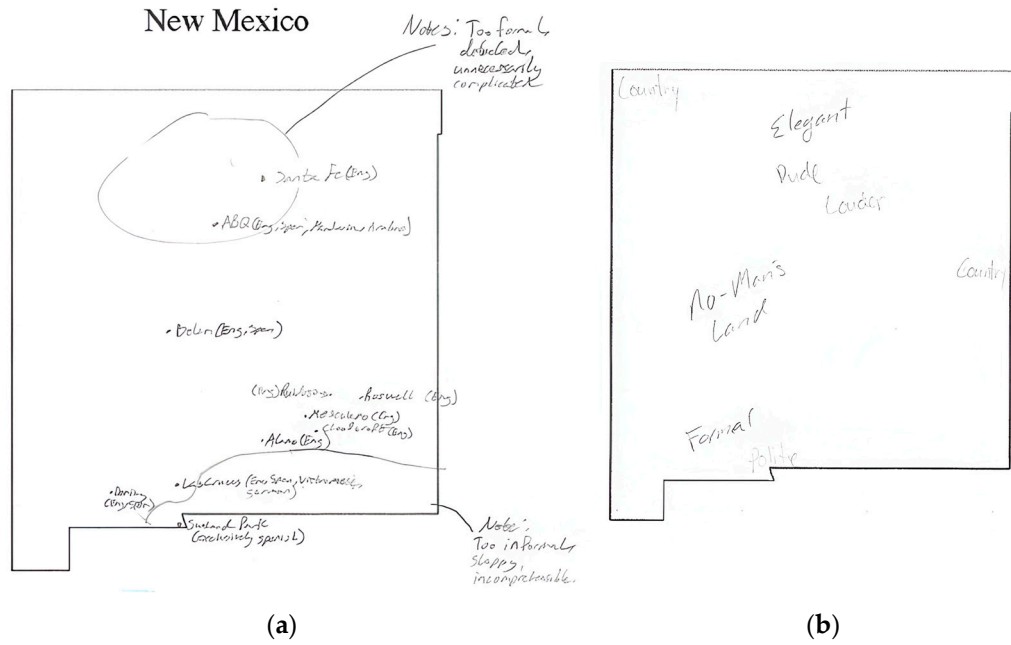

(**a**)                    (**b**)

**Figure 9.** (**a**,**b**) Rural/urban descriptions in New Mexico.

When asked where speakers preferred the way of talking, many people named cities, but this was often cited as a personal connection to the place. For example:

(3)  'I like Las Cruces. I like the diversity in languages. I like that people speak Spanglish like I do.'

Overall, there was a lack of clear juxtaposition (such as Albuquerque is correct/ Hatch is incorrect and filled with slang) that would point to a clear urban/rural divide, but there were some suggestions that it may be evident to some participants. This lack of recognition of rural/urban and incorrectness/correctness, respectively, is a primary difference between the current research and previous work. In comparing these results to those of Northern New Mexico, it is important to note that the south of New Mexico is much more rural than the north, which may be a contributing factor.

### 4.4. Languages: Use, Judgements, and Mixing

One observation that was presented on almost every map was the language spoken in different parts of New Mexico. Overall, many participants (38%) only referenced languages on their maps. Overall, 24% of total maps (*n* = 74) mention only Spanish, English, and/or Spanglish. Many participants referenced different dialects, such as "Texas English" along the east border and "proper" or "midwest" English along the northern border with Colorado (as seen in 1 above).

Figure 10 presents six boxes in which the participant indicates which languages are spoken and with which accents. For example, in Albuquerque, Santa Fe, and Socorro, the

participant indicates that "grammatical English" is spoken, while in Alomogordo, we can see "multiple European English accents".

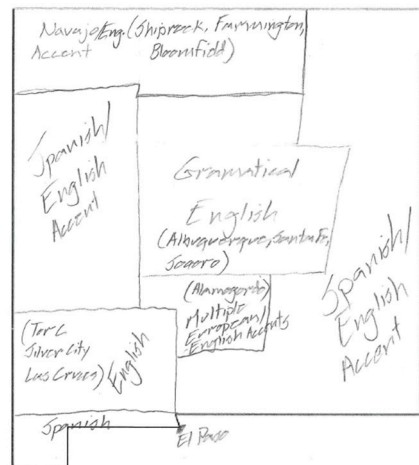

**Figure 10.** Example of language use division.

This participant also references a Navajo English accent in the northwestern part of the state. Some maps focused on more English use, while others discussed Spanish use more. Only 3 maps of the 314 did not mention more than one language explicitly. In fact, many participants differentiated between Spanish/English and Spanglish use, and some even quantified the use of each language as seen in Figure 11a,b.

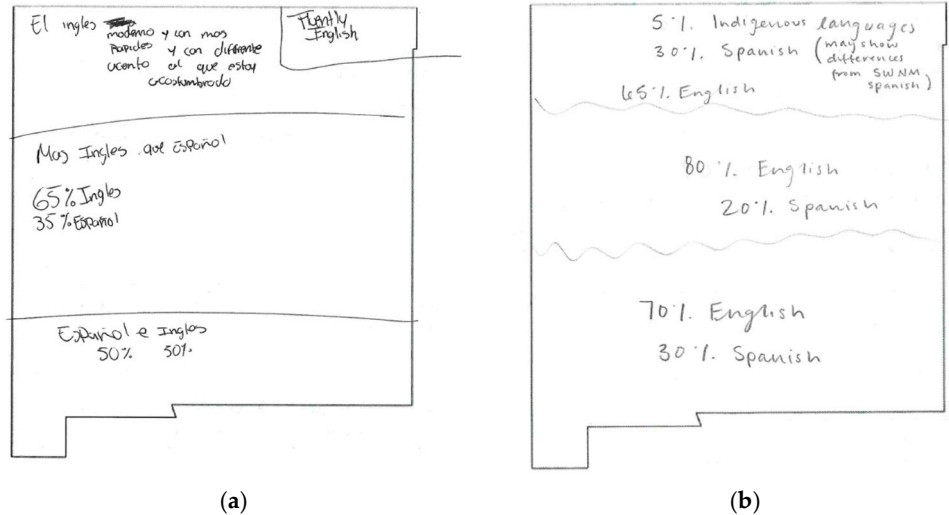

| (a) | (b) |

**Figure 11.** (**a**,**b**) Perceived percentages of languages in New Mexico.

These maps, which also present northern/central/southern zones, both indicate the amount each language is spoken in the area, with the most Spanish spoken in the north and south and more English in the center zone. In Figure 11a, the participant states that *el inglés moderno y con más rapides y con diferente acento al que estoy acosutbrado* "modern English and faster and with a different accent than what I'm used to" (translation mine). In Figure 11b, the participant includes indigenous languages in the northern zone, as well as noting that the Spanish is different than southern NM Spanish. Both maps show more English in the center of the state, which was reflected on many maps. There was also frequent reference to the use of Spanglish, as seen in (3) above. There was evidence of both positive and negative feelings towards Spanglish through the maps and questionnaires, as well as a general identification of where Spanglish is used. However, many people

commented on how the use of Spanglish made them feel comfortable. Vergara Wilson and Koops (2020/2015, p. 183) note that the majority of participants that reference Spanglish allude to the negative aspects of language mixing. There were, however, a few participants that discussed Spanglish as positive in reference to specific cities. This is another difference between southern and northern participants.

One trend observed in the present data that was not discussed in depth in the work of Vergara Wilson and Koops (2020/2015) was the mention of other languages. In total, 45 maps (14%) mention Native people and languages, such as Navajo (Diné), Zuni, Mescalero, Apache, Jicarilla Apache, Hopi, Acoma, and Pueblo.

Figure 12 shows a map in which the participant mentions three native languages in the northern half of the state. The majority of Native languages are written in the northwest corner of the New Mexico maps, but there is some reference in the east to native groups such as the Mescaleros. A few participants also indicated that other languages are spoken here, including Arabic, Russian, Mandarin, and Vietnamese, as seen in Figure 8a. These are all associated with Albuquerque, Santa Fe, or Las Cruces, indicating a perceived connection between urban areas and additional non-Native languages.

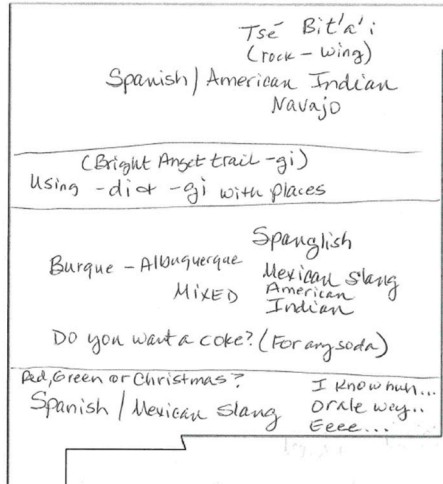

**Figure 12.** Native people on New Mexico maps.

*4.5. Lexical Categories*

Of the maps collected, 105 maps contained information beyond geography and language use. As the instructions were left open-ended on purpose, the content of the maps varied greatly. For example, Figure 13 shows a map that is split into seven categories. Most of these areas include city labels as well as several classifications of language or people. For example, in the southwest, the participant identified Chicano or "pocho" Spanish and "fresa" Spanish in the southwest due to proximity with Juárez, Chihuaua. In the area where Albuquerque and Santa Fe are found, the participant state that there is "Spanish without the spice", and "either you speak bad Spanish or you 'don't' in Las Cruces".

Following Bucholtz et al. (2007), the current discussion will include the following categories of lexical descriptions of speakers: geographic regions/labels, social and linguistic labels, language and dialect labels, slang and other lexical labels, and social group and attitude labels. One of the most frequently used or referenced labels as geographic, based on the north/south divide seen in the current data as well as in the work of Vergara Wilson and Koops (2020/2015). Participants make frequent reference to cardinal directions (as discussed above) as well as reference to the borders, as seen below.

(4)   'English is professional along the Colorado border to the north, "ghetto" in Albuquerque, and people are well spoken again along the Mexican border'.

Here, we see reference to state borders, cities, and the U.S–Mexico border. Throughout the maps, there is minimal reference to any natural landmarks such as mountains or lakes.

There was frequent reference to Native Reservation, such as the Mescaleros in the east and Navajo in the northwest, but these geographic areas directly correspond to languages indicated by participants.

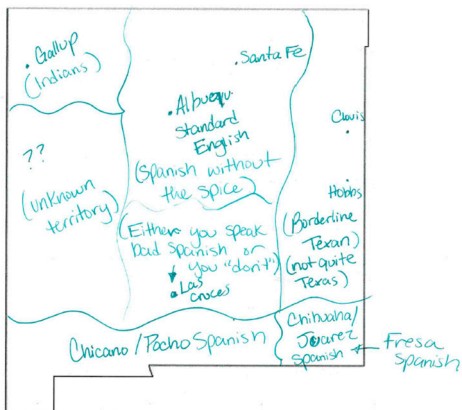

**Figure 13.** Map with a variety of information on New Mexico language use.

Social and linguistic descriptions were frequent and often categorized speakers instead of giving descriptions of language itself. First, New Mexico is known for its chiles, and the state question is 'red or green?' in reference to red or green chile, an important part of New Mexican identity. Several participants included this in language information, indicating that this is important for the language of New Mexico. Additionally, several participants included descriptions of speakers:

(5) North: outgoing, active, proper, polite, harsh
versus
South: small town talk.

(6) North: spoiled, drama, 'oh si'
versus
South: *traviesos* 'mischievous', *chismosos* 'gossipy', *ooooo que la* 'uh oh'.

Both descriptions (as seen in 5) as well as examples (as seen in 6) paint a picture of speakers in New Mexico rather than the language that they produce. These examples of speech are similar to the 'hella' found in Bucholtz et al. (2007). More examples of this will be discussed below.

There were many comments on language use and dialects provided on the maps. First, some comments identified individual dialects found within New Mexico, such as the following:

(7) *Cantadito en el norte y en el sur hay un español mexicano sin acento*. 'Sing songy in the north and in the south there is a Mexican Spanish without an accent'.

In this naming of dialects, the participant makes a clear divide through the middle of the state and indicates that the *cantadito* accent, which is often commonly associated with northern Mexico, is used in the north of New Mexico. Additionally, this participant indicated that the south sounds 'Mexican, but lacks any accent'. There were many descriptions of dialects given throughout the state, such as the following:

(8) 'Proper English next to Colorado, Ghetto English in Albuquerque, Country Spanish in middle, Spanish in south, Spanglish on TX border'.

This participant identified several dialects in almost perfectly parallel layers dividing the state, as seen in Figure 6a,b above. The identified dialects reference language "correctness" with reference to "proper", "country" and "ghetto".

Next, like the linguistic descriptions above, slang, and other lexical labels were able to suggest trends in language perceptions of this area. For example, there was reference to terms such as *tacuachi* (young Mexican–American man who drives big trucks and wears

expensive clothes; literal meaning: possum) and *trokiando*, *cuh* (cruising in big trucks, from *trucking, cousin*), which have a strong connection to Chicano identity and can be found in both English and Spanish discourse in this area. Participants referenced this language in association with more rural parts of the state, suggesting that the use of this language is tied to perceptions of the rural/urban divide. There were also examples of code switching that include the use of *es que* (meaning 'the thing is. . .', literally 'it is that'), *ayay* (used in Mexico to indicate that the previous statement was a joke), and *ay wey* (a colloquial statement of surprise) in both English and Spanish discourse, which speaks to perceived language mixing. There were also some interesting, but perhaps less indicative lexical observations, such as the use of 'no yeah/yeah no' in Albuquerque and 'y'all' in Las Cruces. There were also several references to the use of 'all' throughout the state, such as 'she was all mad'.

Lastly, several participants mentioned social groups within the map. Specifically, participants mentioned *pochos* (pejorative term for people with Mexican background that are not fluent in Spanish), *narcos* (people involved in the drug trade), *fresas* (young Mexican people who came from wealthy, educated backgrounds), and *cholos* (people associated with a Chicano identity that have a certain style), but none of these were specific to one geographic area. Both *pocho* and *fresa* can be seen in Figure 13 above. Participants located all four of these social groups throughout the state, suggesting no consensus on a connection between geography and social groups.

## 5. Conclusions

This study aimed to replicate Vergara Wilson and Koops' (2020/2015) study in Southern New Mexico. Overall, the current findings support much of what Vergara Wilson and Koops (2020/2015) found, but there are a few additional contributions to the body of research. Below, I revisit the research questions:

RQ 1. To what extent do the north/south and rural/urban divides observed in previous work also appear in the current data set?

As Bills and Vigil (2008) describe, the participants of this study reference a north/south divide in both the descriptions of language and the physical dialect boundaries drawn on the maps. This supports Vergara Wilson and Koops' (2020/2015) conclusion about New Mexican speakers. The current study also found that additional north/south layers were common, especially a third central layer that contained the most populated areas of Santa Fe and Albuquerque, which often was identified as mostly English speaking. Evidence of this north/south divide was much clearer than a rural/urban divide. There was not explicit reference to the rural/urban divide as was found in previous studies, but there were several suggestions that participants may observe this trend. Like Fridland and Bartlett (2006) found in their study of Memphis, Tennessee, through was the identification of the urban areas on the map, there was evidence of this distinction, such as references to cities with more "correctness" judgements, and lexical examples such as *trociando, cuh*, which were cited as country or rural.

RQ 2. How do southern New Mexicans perceive their language, and to what extent does this vary across the state?

First, the current data show that New Mexicans in the southern part of the state recognize English and Spanish use and the prevalence of language mixing and Spanglish. All but three participants mentioned at least Spanish and English on their maps, and many assigned language use to each area. The most frequent observation is more Spanish in the north and south with a central zone that is predominantly English. Many speakers communicate their pride in the way that the language is mixed in the southern part of the state, as well as the presence of Mexican identity along the border. There is frequent reference to both U.S.–Mexico and state borders of Colorado, Arizona, and Texas throughout participant responses, following Cramer's (2013) finding of border as an identity marker for participants. However, there seems to be less of a consensus on where this correctness is located than in the study by Vergara Wilson and Koops (2020/2015). There were also

many other languages mentioned, which was not indicated in Vergara Wilson and Koops' (2020/2015) study. These languages included Native languages as well as languages of immigrant communities in the larger cities.

RQ 3. What language do participants use to divide language groups (i.e., words, prosody, food, descriptions of people, etc.)?

Language use was a very common theme in this data set, but there were other interesting markers of identity that participants identified, such as geographic regions/labels, social and linguistic labels, language and dialect labels, slang and other lexical labels, and social group and attitude labels. These data may seem light and fun, but they can also shed light on possible perceptions of New Mexicans. For example, there was extensive reference to borders and the north/south of the state, which highlights the importance of these concepts in New Mexican identity. Similarly, we see labels such as "country", "chicano", "Texas talk", and "cantadito" that describe the accents and make important distinctions in speaker groups. We also see that green and red chile are such an important part of New Mexican identity that the concept appears several times on the maps when discussing language.

The current study aimed to provide an overview of linguistic perception in New Mexico and compare a southern perspective to previous work conducted in Northern New Mexico. While this discussion was able to provide insight into these perceptions, there are several ways in which these data can be expanded in the future. First, this qualitative study did not analyze all sociolinguistic data that were collected. Second, this analysis was impressionistic, identifying patterns in responses. In the future, we are planning on building upon this analysis with a collaboration for a large GIS analysis of all New Mexico data (both Northern and Southern). This will further highlight trends that could shed light on New Mexico language perceptions.

**Funding:** This research received no external funding.

**Institutional Review Board Statement:** This study was conducted in accordance with the Declaration of Helsinki, and approved by the Institutional Review Board (or Ethics Committee) of New Mexico State University (protocol code 19262 and date of approval 1 April 2020).

**Informed Consent Statement:** Informed consent was obtained from all subjects involved in the study.

**Data Availability Statement:** All data reported in results are archived and can be viewed by contacting PI.

**Acknowledgments:** Special thanks to all of the student data collectors from SPAN340: Introduction to Linguistics, as well as to Damián Wilson Vergara and Mark Waltermire for helpful conversations and the two anonymous reviewers for their very helpful comments.

**Conflicts of Interest:** The author declares no conflicts of interest.

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
