# Peer review of "Language Perceptions of New Mexico: A Focus on the NM Borderland"

_languages, doi:10.3390/languages9050161_

Round 1

Reviewer 1 Report

Comments and Suggestions for Authors

This is a wonderful paper and was a pleasure to read. I'm excited to see it published. A few minor suggestions follow. 

--Attend to the spelling/presentation of author names: Is it Vergara Wilson & Koops or Loops & Vergara Wilson? Is it Niedzielski or Niedzielsky?

--Make sure to arrange that the editors of the journal obtain images that are readable. I could not make out anything in most of the maps pictured.  

--Bucholtz et al in their study of 'best' and 'worst' in CA recognize that when university researchers present participants with surveys that include these terms, they may be implicitly endorsing a standard language ideology. I would suggest acknowledging this while also highlighting the value of doing so. I believe that Vergara Wilson and Koops touches upon this topic of 'best/worst'.

--Methods: The author needs to give a little bit more information on methods. It seems as though they are drawing data from both maps and surveys. Also, the author does not really explain how they analyzed the data. It is fine to say that they conducted a thematic evaluation of responses to 'best/worst' that are bolstered by the maps. Or is it vice versa? Either way, please the methods more clearly. Also, did participants have a choice as to what language they took the survey in? If they were not offered in Spanish or English, this would mean a slight departure from Vergara Wilson and Koops. 

--Walk the reader through the maps. The manuscript presents many maps and then talks about them as though the reader understands what the author implies. Highlight examples from the map in the text and talk about why they are important, as with the survey data. 

--The conclusion. This is a great paper! The last paragraph that proposes future directions almost seems too critical. I would suggest to put future directions in its own subheader to signal to the reader that these are thoughts that stand apart from the conclusion. Specifically, do not say, "Second, analysis was very impressionistic, identifying patterns in responses". The paper is fine and the author does not need to offer this self-effacing criticism. The part about wanting to do a quantitative analysis is almost too general and does not give enough info beyond just looking at language preference. What would preference correlate with? Rework this part, please. 

Comments on the Quality of English Language

See above in overall suggestions. But please read for minor typos and such. 

Author Response

Language Review:

Revisions- Reviewer 1

Thank you so much for your helpful review. I have addressed each comment below, and I am attaching the updated manuscript with all changes marked in red font. Thank you for your help!

Reviewer Comment

My Response

--Attend to the spelling/presentation of author names: Is it Vergara Wilson & Koops or Loops & Vergara Wilson? Is it Niedzielski or Niedzielsky?

This has been corrected, thank you.

--Make sure to arrange that the editors of the journal obtain images that are readable. I could not make out anything in most of the maps pictured.  

The map images were updated.

--Bucholtz et al in their study of 'best' and 'worst' in CA recognize that when university researchers present participants with surveys that include these terms, they may be implicitly endorsing a standard language ideology. I would suggest acknowledging this while also highlighting the value of doing so. I believe that Vergara Wilson and Koops touches upon this topic of 'best/worst'.

I have added this to the lit review and in the methods section, I have added the following: Next, participants filled out a survey in which they were also asked about their own language use and preferences. In an effort to replicate the study done by Vergara Wilson and Koops (2020/2015), I wanted to include questions that ask the “best” and “worst” language, but as previous studies like Bucholtz et al. (2007) recognize that the use of these terms may be implicitly endorsing a standard language ideology, the questions were reworded to ask “In your opinion, in which part of the state do you like the way people speak? Why?”. This is a slight variation of the “best”/”worst” question used in previous research.

--Methods: The author needs to give a little bit more information on methods. It seems as though they are drawing data from both maps and surveys. Also, the author does not really explain how they analyzed the data. It is fine to say that they conducted a thematic evaluation of responses to 'best/worst' that are bolstered by the maps. Or is it vice versa? Either way, please the methods more clearly. Also, did participants have a choice as to what language they took the survey in? If they were not offered in Spanish or English, this would mean a slight departure from Vergara Wilson and Koops. 

I have added the following paragraph to the methodology:

 The current study focuses primarily on the data provided on the maps by participants. In the analysis of these maps, there were several considerations. First, the number, placement, and shape of any boundaries such as lines, as well as discussion of borders, cities, and natural landmarks. Second, I noted descriptions of language, which included reference to specific languages, descriptions of dialects, or other language judgements. Additionally, based on Vergara Wilson and Koops (2020/2015), I noted reference to “north/south” and descriptions of rural/urban areas. Finally, lexical descriptions were noted. I follow Bucholtz et al. (2007) in using the following labels to categorize lexical descriptions of speakers: geographic regions/labels, social and linguistic labels, language and dialect labels, slang and other lexical labels, and social group and attitude labels. For the current study, only the “best/worst” questions were considered and used to support the data collected from the maps.

--Walk the reader through the maps. The manuscript presents many maps and then talks about them as though the reader understands what the author implies. Highlight examples from the map in the text and talk about why they are important, as with the survey data.

Thank you for this suggestion. I have added some descriptions and highlights to each map.

--The conclusion. This is a great paper! The last paragraph that proposes future directions almost seems too critical. I would suggest to put future directions in its own subheader to signal to the reader that these are thoughts that stand apart from the conclusion. Specifically, do not say, "Second, analysis was very impressionistic, identifying patterns in responses". The paper is fine and the author does not need to offer this self-effacing criticism. The part about wanting to do a quantitative analysis is almost too general and does not give enough info beyond just looking at language preference. What would preference correlate with? Rework this part, please. 

Thank you! The part about future quantitative analysis has been removed. I mention future GIS expansion of this research, and I agree with the reviewer that this should be enough.

Reviewer 2 Report

Comments and Suggestions for Authors

Review of “Language Perceptions of New Mexico: A focus on the NM borderland”

This article is a replication of an earlier study on perceptual dialectology in New Mexico. The author provides data that supports the existence of a perceived north/south divide but pays more attention to the perceptions of people in southern New Mexico. The abstract and introduction are less clear on what aspects of “how New Mexicans see language in their state” than one would hope, and I have given many comments below on how to adjust to as to attune to reader expectations. A lot of material is included in the background research, but it is not only unclear but also excessive in places where it need not be. For instance, the author includes detailed descriptions of older studies with information that does not seem like it could have any bearing on what they have done with their own research. Much of the literature review reads like items the author happened to have read. It also does not provide the relevant context needed to understand the point of the research. The methods are described succinctly, but, as with the literature review, I found it lacking in detail. The discussion of results for the maps is also very minimal; we have no idea what makes a “trend”, there are references to other studies with similar findings but minimal engagement, and the connection back to Vergara Wilson and Koops is not clear or consistent. We get more ideas of what is meant by “trend” in section 4.2 with the inclusion of percentages, and having more data like this might be helpful in other parts of the results section. There is also a lot of good qualitative data that needs to be elaborated on. Finally, in addition to these concerns, I found the entire paper to contain a number of typos and other errors that I would have expected to be caught ahead of submission.

Please see below for specific comments (by line, using the PDF that was supplied to me for this review). These, in addition to the broader concerns, should be addressed before reconsideration.

·         Lines 7-8: Did they only focus on Northern New Mexicans? The next sentence (lines 8-9) says it is a replication in the South. This is certainly explained more clearly later in the article; just needs to be clearer in the abstract as well.

·         Lines 19-22: I’ll admit this may have been my own bias that got in the way, but I was surprised to see that the focus was on Spanish after having read the abstract. At least in mentioning previous work (Bills and Vigil), you will want to make it clear that the north/south thing is Spanish.

·         Line 23: “…northern part of the state” but you said they worked on New Mexico and Southern Colorado. Needs to be clearer.

·         Line 25: I am unfamiliar with “transfronterizo life”; maybe a source/citation or definition?

·         Line 26: “perceptional” should be “perceptual” (and maybe include a citation too)

·         Lines 29-34: The field has been around longer and typically focuses on the whole country. Your citations here are special in that they focus on single states/cities, like you are. So you will want to make this distinction clear.

·         Line 47: This is the first mention of “rural/urban” and needs to be mentioned when you discuss the background info in the previous paragraph.

·         Line 49: I was also not ready for “lexical categories” as a research question. In terms of structure, it is best if the text preceding them introduced the ideas you are questioning in as clear ways as possible. You go into greater detail later, but you need something to preface the questions in the intro as well.

·         Lines 51-54: Unless you really need the roadmap, it doesn’t add much. It would be much better if you hint to the results here and provide the answer to the “so, what?” question readers may have.

·         Line 57: What is meant by “opinions and beliefs of language”? Beliefs about how language works? A clear definition of “folk linguistics” and its connection to perceptual dialectology is crucial.

·         Line 67: It’s a more positive approach to simply state the uses Albury and Martinez give and leave out the negative ideas of what it is/does. Preston also has a number of justifications for the field.

·         Lines 71-72: “and legitimizes communities underrepresented language.” Should “communities” be possessive?

·         Lines 72-73: I believe theorizing about the impact of the border could be important for your article. I’m not sure where it would go, exactly, but a section that handles the idea of the border could be good. Even non-linguistic work, like Qué onda? By Cynthia Bejarano. Maybe with/after the section on the research area?

·         Starting at line 78: This whole paragraph is just a list of things people have done in perceptual dialectology. I’m afraid readers will not be interested. You can probably simply say that there are many methods but the most common is the draw-a-map task, then move into the part where you describe that task specifically.

·         Line 96: Before you say what they generally draw, you have to clearly describe what they are typically asked to do in this task.

·         Line 97: I think there is something off on this wording: “and to the areas where they consider themselves local”

·         Lines 98-99: These look like your notes: “Preston’s perceptions of the U.S. (a) draw dialect boundaries, (b) label dialect regions, and (c) describe dialect regions.”

·         Line 101: Niedzielski is spelled wrong. (Check the spelling throughout.)

·         Line 108: You’ll need a better-quality image here. This is unreadable. (Same for other figures throughout.)

·         Line 134: “the same rural/urban divide” = same as Fridland and Bartlett? Or something else? You want to explore this in the data you have, but the reader is not set up to understand how this applies to your data at this point. A section on rurality may be useful.

·         Line 143: What makes this “lastly”? You have not covered all research of this type. Much of the literature review thus far reads like a list of things you read, which is not a good way to make an argument. You’ll want to connect each piece together to tell a story.

·         Starting with line 150: If a reader is unfamiliar with New Mexico, this section does not provide enough information or context for them to understand what is important. It may take a lot of space, but you need to describe why we should care about New Mexico, what role Spanish has in the state (and, frankly, why someone like me might have assumed you were talking about English = set New Mexico apart from other US states), and why the concepts of identity, borders, grammatical/lexical features, rurality, etc. are worth our time in this paper.

·         Line 183: This was a nice summary of what appear to be the most important background literature you have included. You need to make sure you connect these results with what you are doing, even if it is a generic transition sentence at the end that says “and I’m going to do this with my data.”

·         Starting at line 185: This is a clear description of your methods, but you need more. You can include the map you gave, provide the survey questions as a list, and maybe describe further who these participants are. How many maps? How did you determine trends? If you describe the methods of Vergara Wilson and Koops here, you can easily say how your task does or does not align.

·         Line 187: “test” should be “task”

·         Line 190: Throughout the article, you cite “Vergara Wilson and Koops”, but here you put Koops first?

·         Line 206: You will need scans of your data images. (Check this throughout.)

·         Line 211: Should be Figure 3.

·         Line 218: “rnorth” should be “north”

·         Line 225: Is this the right figure number?

·         Line 234: Maybe “serves” should be inserted before “as”. I also wonder if you could engage with the connection between your data and what Cramer found.

·         Starting at line 261: It is interesting that the three largest cities do not always get evaluated as best. This differs from other perceptual dialectology findings, and I think it would be important to explicitly show that. Cite other research and say your data don’t do that.

·         Line 284: It is important to note that southern New Mexico is more rural. You should do it earlier in the article. I think a section devoted to “Why (southern) New Mexico?” – with all of the relevant demographic, sociohistorical, geopolitical information would be useful to readers. It could come before the literature review.

·         Line 293: Journals often prefer you to not use “below” because they will put the figures where they need to. You should always use in text references to the figure instead.

·         Line 299: I am curious about the organization of the results. Since the north/south divide section discusses differences in languages (Spanish vs. English, for example), maybe this section needs to come earlier? You might consider the order of the original RQs and use that as a guide to order the results section as a whole.

·         Lines 300-301: What does this mean? “One concept that was presented on almost every map was the language spoken in 300 different parts of New Mexico.” Why “concept”?

·         Line 318: You say there was “no significant reference” but no statistics are provided. I assume you are using “significant” in the general way, but in an article like this, your audience may expect the statistical meaning. There are other places in the article where you use this term or “correlation” when there are no numbers to back it up. Either include the numbers (=do stats) or don’t word it this way.

·         Line 345: This information about following Bucholtz et al. should be presented in the methods section.

·         Line 358: Why “entertaining”?

·         Starting at line 365: These are interesting data points. You need to explore them further. This is just a list.

·         Line 370: What does “this” refer to in “this second speaker”?

·         Line 395: What do these words mean? I can’t understand the “hint” to perceptions without more context. Same with the following bit on codeswitching. And the last section on social groups.

·         Line 411: I like the layout where you reiterate the RQs with what you think you’ve added/shown.

·         Starting at line 461: I appreciate the inclusion of future directions.

·         Line 517: What is the rest of the info for this citation? Also, you cite it as “Vergara Wilson and Koops” elsewhere. And why does it include two years?

·         Line 530: Waltermire forthcoming is missing. [I did not do a full assessment of what is or is not missing and suggest the author do such an assessment.]

Comments on the Quality of English Language

Comments on language issues can be found above. Overall intelligible, just has a lot of typographical and stylistic issues.

Author Response

Language Review:

Revisions- Reviewer 2

Thank you very much for your helpful review. I have addressed each comment in the table below, and I am attaching the updated manuscript with changes indicated in red font. Your comments were very helpful, and I appreciate the time you took to help me improve this manuscript.

Reviewer Comment

My response/actions:

This article is a replication of an earlier study on perceptual dialectology in New Mexico. The author provides data that supports the existence of a perceived north/south divide but pays more attention to the perceptions of people in southern New Mexico. The abstract and introduction are less clear on what aspects of “how New Mexicans see language in their state” than one would hope, and I have given many comments below on how to adjust to as to attune to reader expectations. A lot of material is included in the background research, but it is not only unclear but also excessive in places where it need not be. For instance, the author includes detailed descriptions of older studies with information that does not seem like it could have any bearing on what they have done with their own research. Much of the literature review reads like items the author happened to have read. It also does not provide the relevant context needed to understand the point of the research. The methods are described succinctly, but, as with the literature review, I found it lacking in detail. The discussion of results for the maps is also very minimal; we have no idea what makes a “trend”, there are references to other studies with similar findings but minimal engagement, and the connection back to Vergara Wilson and Koops is not clear or consistent. We get more ideas of what is meant by “trend” in section 4.2 with the inclusion of percentages, and having more data like this might be helpful in other parts of the results section. There is also a lot of good qualitative data that needs to be elaborated on. Finally, in addition to these concerns, I found the entire paper to contain a number of typos and other errors that I would have expected to be caught ahead of submission.

I have responded to each VERY HELPFUL comment below, but overall, I reorganized several sections, added details to others, and hopefully improved the overall analysis. Thank you for your very helpful feedback. All of my changes can be seen in the manuscript with red font color as well.

Lines 7-8: Did they only focus on Northern New Mexicans? The next sentence (lines 8-9) says it is a replication in the South. This is certainly explained more clearly later in the article; just needs to be clearer in the abstract as well.

This has been changed: to study how New Mexicans in the northern part of the state perceive Language of New Mexico.

 Lines 19-22: I’ll admit this may have been my own bias that got in the way, but I was surprised to see that the focus was on Spanish after having read the abstract. At least in mentioning previous work (Bills and Vigil), you will want to make it clear that the north/south thing is Spanish.

I added another sentence to the beginning of the abstract to make this clearer for other readers:

Spanish has a long history in the state of New Mexico, and it significantly contributes to New Mexican identity.

 Line 23: “…northern part of the state” but you said they worked on New Mexico and Southern Colorado. Needs to be clearer.

Changed “the state” to “New Mexico”:

describe a spatial divide between the northern part of New Mexico

Line 25: I am unfamiliar with “transfronterizo life”; maybe a source/citation or definition?

Clarification added: transfronterizo life, in which residents of this area cross the border regularly for work, recreation, family, etc

Line 26: “perceptional” should be “perceptual” (and maybe include a citation too)

Fixed, thank you!

Also, citation added: (introduced by Preston 1989)

Lines 29-34: The field has been around longer and typically focuses on the whole country. Your citations here are special in that they focus on single states/cities, like you are. So you will want to make this distinction clear.

This has been changed:

In recent years, it has been used to study language in individual areas within the United States,

Line 47: This is the first mention of “rural/urban” and needs to be mentioned when you discuss the background info in the previous paragraph.

The following was added to the previous paragraph:

Like previous studies mentioned above, there is also discussion of perception of a rural/urban divide within the state.

Line 49: I was also not ready for “lexical categories” as a research question. In terms of structure, it is best if the text preceding them introduced the ideas you are questioning in as clear ways as possible. You go into greater detail later, but you need something to preface the questions in the intro as well.

“Lexical categories” was removed, and the updated question is as follows:

What language do participants use to divide language groups (i.e. words, prosody, food, descriptions of people, etc)?

Line 57: What is meant by “opinions and beliefs of language”? Beliefs about how language works? A clear definition of “folk linguistics” and its connection to perceptual dialectology is crucial.

The first two sentences in this paragraph were condensed to be able to move the reader quicker to the connection below.

Line 67: It’s a more positive approach to simply state the uses Albury and Martinez give and leave out the negative ideas of what it is/does. Preston also has a number of justifications for the field.

Thank you. I do think it is important to acknowledge the arguments that exist against this methodology.

Lines 71-72: “and legitimizes communities underrepresented language.” Should “communities” be possessive?

communities and underrepresented language.

Lines 72-73: I believe theorizing about the impact of the border could be important for your article. I’m not sure where it would go, exactly, but a section that handles the idea of the border could be good. Even non-linguistic work, like Qué onda? By Cynthia Bejarano. Maybe with/after the section on the research area?

I have added information on the border and identity in the section on background in NM

Starting at line 78: This whole paragraph is just a list of things people have done in perceptual dialectology. I’m afraid readers will not be interested. You can probably simply say that there are many methods but the most common is the draw-a-map task, then move into the part where you describe that task specifically.

This was reorganized to shorten:

More recent additions to the methodology include dialect identification (Cukor-Ávila et al. 2012 in Texas), matched guise tests to measure language attitudes (Garrett, et al. 2003), naming dialects (Alfaraz 2002, 2014 in Miami), imitation (Adank et al. 2013 in Glasgow), and term frequency using social media and heat maps (Garzon 2017 and Callesano 2020 in Miami). Preston (2018) also argues for benefits of using technology such as GIS in modern perceptual dialectology.

  Line 96: Before you say what they generally draw, you have to clearly describe what they are typically asked to do in this task.

The following was added:

Within this methodology, participants are given little instruction other than to indicate how people speak and where.

 Line 97: I think there is something off on this wording: “and to the areas where they consider themselves local”

Yes, the “to” was removed.

Lines 98-99: These look like your notes: “Preston’s perceptions of the U.S. (a) draw dialect boundaries, (b) label dialect regions, and (c) describe dialect regions.”

This sentence was removed.

 Line 101: Niedzielski is spelled wrong. (Check the spelling throughout.)

Corrected, thank you.

 Line 108: You’ll need a better-quality image here. This is unreadable. (Same for other figures throughout.)

Images are updated

Line 134: “the same rural/urban divide” = same as Fridland and Bartlett? Or something else? You want to explore this in the data you have, but the reader is not set up to understand how this applies to your data at this point. A section on rurality may be useful.

The following was added:

a similar rural/urban divide as observed in Fridland and Bartlett (2006)’s Tennesee work.

Line 143: What makes this “lastly”? You have not covered all research of this type. Much of the literature review thus far reads like a list of things you read, which is not a good way to make an argument. You’ll want to connect each piece together to tell a story.

Lastly was deleted.

I have reorganized this section by theme: first urban rural, then differentness, then lexical lists. Each theme presents a couple of studies. I agree with the reviewer that this flows much better. Thank you.

Starting with line 150: If a reader is unfamiliar with New Mexico, this section does not provide enough information or context for them to understand what is important. It may take a lot of space, but you need to describe why we should care about New Mexico, what role Spanish has in the state (and, frankly, why someone like me might have assumed you were talking about English = set New Mexico apart from other US states), and why the concepts of identity, borders, grammatical/lexical features, rurality, etc. are worth our time in this paper.

 Line 183: This was a nice summary of what appear to be the most important background literature you have included. You need to make sure you connect these results with what you are doing, even if it is a generic transition sentence at the end that says “and I’m going to do this with my data.”

Thank you. I have added:

The current study will investigate these themes using participants from the southern part of the state.

Starting at line 185: This is a clear description of your methods, but you need more. You can include the map you gave, provide the survey questions as a list, and maybe describe further who these participants are. How many maps? How did you determine trends? If you describe the methods of Vergara Wilson and Koops here, you can easily say how your task does or does not align.

I made several additions to the methdology section (indicated in red on manuscript), including the following paragraph:

The current study focuses primarily on the data provided on the maps by participants. In the analysis of these maps, there were several considerations. First, the number, placement, and shape of any boundaries such as lines, as well as discussion of borders, cities, and natural landmarks. Second, I noted descriptions of language, which included reference to specific languages, descriptions of dialects, or other language judgements. Additionally, based on Vergara Wilson and Koops (2020/2015), I noted reference to “north/south” and descriptions of rural/urban areas. Finally, lexical descriptions were noted. I follow Bucholtz et al. (2007) in using the following labels to categorize lexical descriptions of speakers: geographic regions/labels, social and linguistic labels, language and dialect labels, slang and other lexical labels, and social group and attitude labels. For the current study, only the “best/worst” questions were considered and used to support the data collected from the maps.

Line 187: “test” should be “task”

Thank you, this has been changed.

 Line 190: Throughout the article, you cite “Vergara Wilson and Koops”, but here you put Koops first?

This was corrected.

Line 206: You will need scans of your data images. (Check this throughout.)

Images are updated

Line 211: Should be Figure 3.

This has been fixed

Line 218: “rnorth” should be “north”

Thank you, this has been changed.

Line 225: Is this the right figure number?

This has been fixed (it is 5)

Line 234: Maybe “serves” should be inserted before “as”. I also wonder if you could engage with the connection between your data and what Cramer found.

“Serves” as been added here.

Also, specifically Kentucky’s border and its importance in “southern” identity.

Starting at line 261: It is interesting that the three largest cities do not always get evaluated as best. This differs from other perceptual dialectology findings, and I think it would be important to explicitly show that. Cite other research and say your data don’t do that.

I have added several sentences throughout this section, indicated in red in the document. I reference other work and explicitly state that these data differ from previous work.

Line 284: It is important to note that southern New Mexico is more rural. You should do it earlier in the article. I think a section devoted to “Why (southern) New Mexico?” – with all of the relevant demographic, sociohistorical, geopolitical information would be useful to readers. It could come before the literature review.

Thank you. I have added more about southern New Mexico in the literature review.

Line 293: Journals often prefer you to not use “below” because they will put the figures where they need to. You should always use in text references to the figure instead.

Thank you. This has been changed to “figure 8”

Line 299: I am curious about the organization of the results. Since the north/south divide section discusses differences in languages (Spanish vs. English, for example), maybe this section needs to come earlier? You might consider the order of the original RQs and use that as a guide to order the results section as a whole.

Thank you for this suggestion. I have reorganized both the results and the RQs, and I believe it makes more sense.

Lines 300-301: What does this mean? “One concept that was presented on almost every map was the language spoken in 300 different parts of New Mexico.” Why “concept”?

I have changed “concept” for “observation”

Line 318: You say there was “no significant reference” but no statistics are provided. I assume you are using “significant” in the general way, but in an article like this, your audience may expect the statistical meaning. There are other places in the article where you use this term or “correlation” when there are no numbers to back it up. Either include the numbers (=do stats) or don’t word it this way.

I have replaced “correclation” with “connection” and “significant” with “frequent” as there are no numbers.

 Line 345: This information about following Bucholtz et al. should be presented in the methods section.

The following was added to the methods:

In the analysis of the maps, I follow Bucholtz et al. (2007) in using the following labels to categorize lexical descriptions of speakers: geographic regions/labels, social and linguistic labels, language and dialect labels, slang and other lexical labels, and social group and attitude labels.

 Line 358: Why “entertaining”?

This was deleted.

Starting at line 365: These are interesting data points. You need to explore them further. This is just a list.

Thank you. More text has been added to this section.

Line 370: What does “this” refer to in “this second speaker”?

I’m not sure. I cut this.

Line 395: What do these words mean? I can’t understand the “hint” to perceptions without more context. Same with the following bit on codeswitching. And the last section on social groups.

rural speech known for language mixing with the terms tacuachi (young Mexican-American man who drives big trucks and wears expensive clothes, literal meaning: possum) and trokiando, cuh (cruising in big trucks, from trucking, cousin),

As well as the last two sections

Line 411: I like the layout where you reiterate the RQs with what you think you’ve added/shown.

Thank you.

 Starting at line 461: I appreciate the inclusion of future directions.

Thank you.

 Line 517: What is the rest of the info for this citation? Also, you cite it as “Vergara Wilson and Koops” elsewhere. And why does it include two years?

This was an error. The two dates are included per the journal citation and authors’ request.

Line 530: Waltermire forthcoming is missing. [I did not do a full assessment of what is or is not missing and suggest the author do such an assessment.]

This has been fixed.

Round 2

Reviewer 2 Report

Comments and Suggestions for Authors

This is a fantastic revision! Good work!